# Developing *In Vitro* Models to Define the Role of Direct Mitochondrial Toxicity in Frequently Reported Drug-Induced Rhabdomyolysis

**DOI:** 10.3390/biomedicines11051485

**Published:** 2023-05-19

**Authors:** Faten F. Bin Dayel, Ana Alfirevic, Amy E. Chadwick

**Affiliations:** Department of Pharmacology and Therapeutics, University of Liverpool, Ashton Street, Liverpool L69 3GE, UK; ana.alfirevic@liverpool.ac.uk (A.A.); aemercer@liverpool.ac.uk (A.E.C.)

**Keywords:** FAERS, suspect drug-induced rhabdomyolysis, L6, HSKMDC, skeletal muscle toxicity, mitochondrial dysfunction

## Abstract

The United States Food and Drug Administration Adverse Event Reporting System (FAERS) logged 27,140 rhabdomyolysis cases from 2004 to 31 March 2020. We used FAERS to identify 14 drugs frequently reported in 6583 rhabdomyolysis cases and to investigate whether mitochondrial toxicity is a common pathway of drug-induced rhabdomyolysis by these drugs. Preliminary screening for mitochondrial toxicity was performed using the acute metabolic switch assay, which is adapted here for use in murine L6 cells. Fenofibrate, risperidone, pregabalin, propofol, and simvastatin lactone drugs were identified as mitotoxic and underwent further investigation, using real-time respirometry (Seahorse Technology) to provide more detail on the mechanism of mitochondrial-induced toxicity. To confirm the human relevance of the findings, fenofibrate and risperidone were evaluated in primary human skeletal muscle-derived cells (HSKMDC), using the acute metabolic switch assay and real-time respirometry, which confirmed this designation, although the toxic effects on the mitochondria were more pronounced in HSKMDC. Overall, these studies demonstrate that the L6 model of acute modification may find utility as an initial, cost-effective screen for identifying potential myotoxicants with relevance to humans and, importantly, that drug-induced mitochondrial dysfunction may be a common mechanism shared by some drugs that induce myotoxicity.

## 1. Introduction

Drug-induced rhabdomyolysis is a rare adverse event that is considered the most serious clinical form of skeletal muscle toxicity. Rhabdomyolysis has the potential to cause life-threatening outcomes, resulting in hospitalisation and death [1,2]. Clinically, according to the American College of Cardiology, American Heart Association, and National Heart, Lung and Blood Institute (ACC/AHA/NHLBI), rhabdomyolysis is associated with severe muscle pain accompanied by weakness, myoglobinuria (dark brown urine), and an elevation of serum creatine kinase (CK) to ≥10 × upper limit of normal (ULN) [3,4]; while an FDA Drug Safety Communication defined rhabdomyolysis as unexplained muscle weakness or pain with serum CK > 40 times ULN. Physiologically, it is defined as the rapid destruction of skeletal muscle, with released components leading to further severe complications including electrolyte disturbance, liver failure, and acute kidney injury as a result of the accumulation of myoglobin in the renal tubules [1,2,5,6,7,8,9]. The global incidence of rhabdomyolysis is unknown, but it is reported to account for 7–10% of all acute kidney injuries [10,11]. It can occur in patients without prior muscular disease when exposed to therapeutic doses of certain drugs, but it is more pronounced in patients with predisposing factors including, old age, concomitant diseases, and multiple medications [12,13,14,15].

Drug-induced skeletal muscle injury is well-described, with the sensitivity of muscle attributed to its high metabolic rate and plentiful blood supply [16]. Rhabdomyolysis is the most severe form of muscle toxicity, which often requires the hospitalization of those affected and can be fatal due to subsequent acute kidney failure (AKF). It has been estimated that approximately 10–40% of rhabdomyolysis cases lead to AKF [17,18,19]. Therefore, this issue represents a large burden for the healthcare system, and a large cost for the pharmaceutical industry if the culprit drug has to be taken off the market. Economic costs to the pharmaceutical industry include the cost to develop the therapeutic, currently calculated to range between USD 161 million and 4.54 billion per approved drug [20], as well as the subsequent impact on the company’s value and loss of anticipated sales. For example, the lipid-lowering drug cerivastatin was withdrawn from the market due to increased reports of rhabdomyolysis [21]. It is estimated that this cost Bayer A. G. hundreds of millions of US dollars due to a loss of anticipated sales and subsequent decreases in share value [22]. In addition, it has been widely reported by news outlets that USD 1.1 billion has been paid in out-of-court settlements to individuals affected by this side effect. It has been estimated that if cerivastatin had been withdrawn earlier, then USD 10 million in direct medical costs could have been saved [23]. Ultimately, these costs could have been eliminated if this risk had been detected during drug development. However, to date, simple models that can be deployed to identify potential myotoxicants, including those that may trigger rhabdomyolysis in some individuals, are lacking during the pre-clinical assessment of drug development candidates.

There are many known mechanisms of drug-induced rhabdomyolysis, with sustained elevations in cytosolic calcium concentration combined with a reduction in myoplasmic ATP contents commonly seen [24,25,26,27,28]. The results from several studies suggest that such negative effects on skeletal muscle are attributable, to varying extents, to detrimental effects on mitochondrial function, which have been observed across many drug classes such as antihyperlipidemic agents [29,30,31,32,33,34,35], antivirals [36,37,38,39], and antibiotics [40].

Therefore, this study aimed to identify drugs that are suspected to induce rhabdomyolysis, post-licensing, and to use these to develop a potential screening method based on mitochondrial toxicity to identify drugs with the potential to cause muscle toxicity in patients. To achieve this, a list of suspect drugs was generated using the United States Food and Drug Administration (FDA) Adverse Event Reporting System (FAERS), a standard public website created to monitor the safety of all approved drugs. In this study, we investigated whether drug-induced mitochondrial dysfunction may be a screenable alert for the potential of a drug to induce skeletal muscle toxicity. Here, we describe the adaptation of the acute metabolic switch screen, first described in a liver cell model, for use in L6 rat-derived skeletal muscle cells before confirmatory testing in more physiologically relevant, human primary skeletal muscle cells using the drugs identified by interrogation of the FAERS database as a test panel.

## 2. Materials and Methods

### 2.1. Materials

L6 myoblast cells were purchased from the American Type Culture Collection (Manassas, VA, USA). Human primary skeletal muscle-derived cells (HSKMDC) (catalogue #SK-1111, Lot #P0100750F) were obtained from Cook Myosite Ltd. (Pittsburgh, PA, USA). All forms of Dulbecco’s Modified Eagle Medium (DMEM), rat tail collagen I, media supplements, and cell culture reagents were from Life Technologies (Paisley, UK). The lactate dehydrogenase cytotoxicity detection kit was from Roche Diagnostics Ltd. (West Sussex, UK). ATP lysis buffer and ATP reagent kit were from Sigma Aldrich (Southampton, UK). Extracellular flux analyser (XF^e^96) consumables and base medium were from Agilent Technologies (Santa Clara, CA, USA). Simvastatin hydroxy acid ammonium salt was from Toronto Research Chemicals (LGC Promochem, Middlesex, UK), and the control drugs propofol and pregabalin were purchased from United States Pharmacopoeia (Staten Island, NY, USA) and Merck (West Point, PA, USA), respectively. All other drugs, reagents, and chemicals were purchased from Sigma Aldrich (Dorset, UK).

### 2.2. Data Source, Case Identification, and Accuracy of Data

Suspect drugs were reported to FAERS from the period 2004 to 31 March 2021 (https://www.fda.gov/drugs/questions-and-answers-fdas-adverse-event-reporting-system-faers/fda-adverse-event-reporting-system-faers-public-dashboard; accessed on 1 April 2021) [41]. Due to the quarterly updating of the database by the FDA, the searches were performed using the term “rhabdomyolysis” on one occasion to ensure the data were extracted at that specified period. All individual files were downloaded and saved from this one timepoint to specify the analysis period used in this study. Further refinement was performed using Microsoft Excel to extract only cases where the suspect drug was reported as a monotherapy, using the generic name of the suspect drugs. To ensure data accuracy, any case reported by a consumer or company underwent further checks to ensure it had not been previously reported by a healthcare provider, by confirming that no duplication of case ID was present. Reported odds ratios (RORs) (Equation (1)) were calculated with 95% confidence intervals (CIs) (Equation (2)), with *a*, *b*, *c*, and *d* defined as *a*: the number of cases with the event of interest (rhabdomyolysis) after using the suspected drug, *b*: the number of cases with all other AERs after using the suspected drug, *c*: the number of cases with the event of interest (rhabdomyolysis) with all other drugs, *d*: the number of cases with all other AERs with all other drugs reported to FAERS.
(1)$$ROR=\frac{a/c}{b/d}=\frac{ad}{bc}$$
(2)$$95\%\;CI=exp\left\{log(\mathrm{ROR})\pm 1.96\sqrt{\frac{1}{a}+\frac{1}{b}+\frac{1}{c}+\frac{1}{d}}\right\}$$

### 2.3. Cell Culture

L6 cells were maintained in complete growth media containing DMEM high-glucose media (glucose 25 mM) supplemented with foetal bovine serum (10% *v*/*v*), _L_-glutamine (4 mM), sodium pyruvate (1 mM), and HEPES (5 mM). HSKMDC was maintained in DMEM low-glucose media (glucose 5 mM), _L_-glutamine (4 mM), sodium pyruvate (1 mM) supplemented with foetal bovine serum (8% *v*/*v*), epidermal growth factor (10 ng/mL), fetuin (50 μg/mL), and dexamethasone (0.4 μg/mL, 1 μM). Although it is reported that dexamethasone may interact with mitochondrial function at nM levels, it is essential in the culture media to induce cell proliferation. However, in mitigation of this, all drug-induced changes are assessed relative to vehicle control. All cells were incubated in humidified air containing 5% CO_2_ at 37 °C. The cells were utilised to passage 12 for L6 and between passages 7 and 10 for HSKMDC.

### 2.4. Cell Plating and Induction of Differentiated Myotubes of L6 and HSKMDC

L6 myoblasts were dissociated when at a confluence of 60–80%, using trypsin-EDTA (0.25%). They were then seeded into collagen-coated wells (50 µg/mL in 0.02 M of acetic acid) at a density of 16,000 cells/100 μL/well (96 well plates) and 10,000 cells/100 μL/well (XF 96-well cell culture microplates). The cells were next incubated for 24 h in complete growth medium to promote adherence, after which the growth medium was removed and the cells washed twice with 1× HBSS. The L6 myoblasts were then cultured in differentiation medium consisting of high-glucose DMEM (25 mM) supplemented with _L_-glutamine (4 mM), sodium pyruvate (1 mM), HEPES (5 mM), and horse serum (2% *v*/*v*). Fresh differentiation medium was provided every 48 h for 7 days. Differentiated myotubes were confirmed through visual inspection under a light microscope.

HSKMDC myoblasts were collected by trypsinisation using trypsin-EDTA (0.05%) upon reaching 50% confluence and seeded into collagen-coated wells, at a density of 20,000 cells/100 μL/well (96 well plates) and 25,000 cells/100 μL/well (XF 96-well cell culture microplates). After 24 h, the cells were washed twice with 1× PBS (-/-), and growth media were replaced with differentiation medium containing DMEM low-glucose media (glucose 5 mM), _L_-glutamine (4 mM), sodium pyruvate (1 mM) supplemented with 2% (*v*/*v*) horse serum, and (10 nM) human insulin. The medium was refreshed every 48 h for 5 days. Differentiated and contractile myotubes were confirmed via visual inspection under a light microscope.

### 2.5. Acute Metabolic Switch on Differentiated Myotubes

On the day of the assay, the media on the differentiated myotubes were replaced with either serum-free high-glucose (DMEM, 25 mM of glucose, 4 mM of l-glutamine, 1 mM of sodium pyruvate, and 5 mM of HEPES) or serum-free galactose (DMEM, 10 mM of galactose, 6 mM of l-glutamine, 1 mM of sodium pyruvate, and 5 mM of HEPES) media (100 µL) for 2 h (37 °C, 5% CO_2_). The glucose and galactose media were removed and replaced with drug-dosing solutions in either glucose or galactose media as appropriate, before incubation (2 h, 37 °C, 5% CO_2_). Drugs were dissolved in DMSO, except for pregabalin (dH_2_O) and ezetimibe (dimethylformamide (DMF)), as per vendor instructions, with the final concentration of diluent at a constant of 0.5% *v*/*v*. Vehicle control was included in each experiment.

### 2.6. Measurement of Cellular ATP Content Following Drug Exposure on L6 and HSKMDC Differentiated Myotubes

The cellular ATP content was measured using a bioluminescence assay kit, alongside the concomitant measurement of protein using a bicinchoninic acid (BCA) assay kit to normalise ATP data to the total protein. Briefly, the cells were lysed using ATP lysis buffer (50 µL/well) and the plate was shaken (5 min). According to the manufacturer’s instructions, 40 µL of ATP reagent was added to each well containing 5 µL of cell lysate in a white 96-well plate. The ATPase luminescence signal was measured immediately using a plate reader (Varioskan, Thermo Scientific, Schaumburg, IL USA). In a separate plate, 200 µL of BCA reagent was added to 10 µL of cell lysate and incubated at 37 °C, 5% CO_2_ for 30 min, before absorbance was measured at 562 nm using the plate reader.

### 2.7. Assessment of Cell Death Using Lactate Dehydrogenase (LDH) Content on (HSKMDC) Differentiated Myotubes

Cytotoxicity was detected using a lactate dehydrogenase (LDH) assay kit. Briefly, according to the manufacturers’ instructions, 25 µL of the supernatant and 10 µL of cell lysate were incubated with 50 µL of LDH reagent for 30 min in the dark. Samples were read at 490 nm using a plate reader.

### 2.8. Mitochondrial Stress Test in Differentiated Myotubes Using Seahorse Flux Analyser (XF^e^96)

To confirm the mitochondrial toxicity of the suspect drugs positive in our first screen on L6 cell lines, an extracellular flux analysis was used. The mitochondrial stress test allowed confirmation of mitotoxicity and further identification of the specific mechanisms through quantification of OXPHOS parameters: basal respiration (BR), ATP-linked respiration (ALR), and spare respiratory capacity (SRC). On the day of the assay, for the analysis of the acute injection of the drug on differentiated myotubes, the differentiation media was replaced with 175 µL of unbuffered Seahorse Assay media supplemented with 25 mM of glucose, 4 mM of _L_-glutamine, 1 mM of sodium pyruvate, and 5 mM of HEPES pre-warmed to 37 °C, then incubated in a CO_2_ free incubator at 37 °C for 1 h. Alternately, in the pre-treatment strategy, differentiated myotubes were pre-treated with the drug for 5 h in differentiation medium, then transferred to a CO_2_-free incubator at 37 °C for 1 h. Next, the differentiation media were replaced with 175 µL of unbuffered Seahorse Assay media supplemented with the constituents stated above. Briefly, using the XF^e^96 instrument (Seahorse extracellular flux analyser, Agilent Technologies, Santa Clara, CA, USA), the oxygen partial pressure in each well was allowed to reach equilibrium for 10 min. The oxygen consumption rate (OCR) was measured three times at baseline; then, there was a 3 min mix for each measurement followed by a 3 min wait. This was followed by an injection of sequential mitochondrial inhibitors of oligomycin (1 µM) (ATP synthase inhibitor), carbonyl cyanide 4-(trifluoromethoxy) phenylhydrazone (FCCP) (0.75 µM) (ionophore), and rotenone/antimycin A (1 µM each) (complex I/III inhibitors, respectively). In the acute injection method, the drug injection was only performed at the end of the basal measurement cycles, followed by the mitochondrial inhibitors described above. The duration of the assay was 120 min (acute injection) and 90 min (pre-treatment strategy). From each mitochondrial stress test was determined the OCR parameters of non-mitochondrial respiration (NMR = lowest measurement after rotenone/antimycin A); basal respiration (BR = last measurement before oligomycin − NMR); proton-leak (PL = BR − lowest respiration after oligomycin); ATP-linked respiration (ALR = BR − PL); maximum respiratory capacity (MRC = highest measurement after FCCP − NMR); and spare respiratory capacity (SRC = MRC − BR) [42,43].

### 2.9. Statistical Analysis

Data are calculated from at least three independent experiments (*n* = 3) with triplicate replicates in each experiment, and all values are expressed as mean ± standard deviation (S.D.). The statistical analyses were performed using GraphPad Prism^®^ version 9 software (GraphPad Software, Inc., San Diego, CA, USA). Data were tested for Gaussian distribution using the Shapiro–Wilk normality test before statistical significance was determined using a Student’s *t*-test with Welch’s correction, one-way, or two-way analysis of variance (ANOVA), with Dunnett’s correction for multiple comparisons. A *p*-value ≤ 0.05 was accepted as significant.

## 3. Results

### 3.1. The Most Common Drugs Suspected to Induce Rhabdomyolysis, as Reported to FAERS, Were Identified

In the FAERS database, there were 27,140 reported rhabdomyolysis cases during the period studied (Table 1). From these, the top 31 drugs were selected based on the criteria that the suspect drugs represent greater than 1% of the total reported rhabdomyolysis cases.

These cases were then further refined to extract those where only a single suspect drug was reported (as monotherapy). The extraction identified 6583 cases, with 14 of the most frequently reported suspect drugs representing ≥1.8% of the total cases (Table 2). Importantly, this excluded all cases where the drug of interest was reported alongside more than one other suspect drug to rule out the possibility of rhabdomyolysis being induced due to drug/drug interactions, which is outside the scope of this investigation. Consistent with previous reports, statins were most frequently reported as a suspected cause of rhabdomyolysis with single use or with other concomitant drugs [45].

### 3.2. The Measurement of Cellular ATP Content Can Detect the Early Onset of Mitochondrial Dysfunction in L6 Cells

The suitability of L6 cells for the acute metabolic switch assay was determined by using the classical mitochondrial toxin rotenone as a positive control (Figure 1) [46]. In this assay, the effect of a test compound or drug on ATP levels is measured in media supplemented with glucose and media supplemented with galactose. Mitotoxic compounds induce a more potent effect when tested in galactose-supplemented media due to the suppression of compensatory glycolytic ATP production. Based upon this test, compounds are labelled as direct mitotoxins if the values for IC_50_-ATPglu and IC_50_-ATPgal are statistically significant (using one-way ANOVA) and, in most cases, this occurs alongside an IC_50_-ATPglu/IC_50_-ATPgal ratio ≥ 2, as previously described [43,47]. Significant differences in IC_50_, below the ratio of 2 (<2), indicate that the mitochondria may be involved as part of a multi-mechanistic pathway. In L6 cells, rotenone (2 h) causes a reduction in ATP production in cells cultured in galactose at significantly lower drug concentrations compared to those in high-glucose medium (Figure 1, Table 3).

The top suspect drugs, identified in Table 2, were next tested in L6 cells using the acute metabolic switch assay (Figure 2, Table 3). In the standard assay, cells were exposed to drugs (0–300 µM). For those with lower solubility a reduced concentration range was tested (0–100 µM). Ezetimibe/Simvastatin was excluded from screening due to limitations in the solubility of the combination. Three of the fourteen compounds tested (fenofibrate, pregabalin, and risperidone) could be classified as containing a mitochondrial liability based upon the IC_50_-ATPglu vs. IC_50_-ATPgal, as defined in Figure 2A and Table 3. Fenofibrate was the most potent of the compounds tested (IC_50_-ATPgal 30.0 ± 19.4 μΜ). Propofol (IC_50_-ATPglu/gal > 1.2) and simvastatin lactone (SVL), the inactive form of simvastatin acid (IC_50_-ATPglu/gal = 1.5), were classified to be multi-mechanistic as they caused differences in IC_50_ below the ratio of 2 (<2), indicating that propofol and SVL could have multifactorial toxicity rather than mitochondrial dysfunction alone (Figure 2B and Table 3). Over this short time point, quetiapine and clozapine were seen to affect cellular ATP content with no statistical difference between the two mediums (IC_50_-ATP glu/gal = 1.2 and 1, respectively). On the other hand, the remaining seven compounds: olanzapine, simvastatin acid, atorvastatin acid, rosuvastatin, ezetimibe, levetiracetam, and daptomycin showed no effect on cellular ATP content in either media (IC_50_-ATP glu/gal = 1) (Figure 2C and Table 3).

### 3.3. Respirometry Reveals That the Suspect Drugs Identified as Mitotoxins Inhibit Electron Transport Chain (ETC) Activity in L6 Cells

Extracellular flux analyser technology (XF^e^96) was used to confirm the mitotoxic effects of fenofibrate, pregabalin, risperidone, propofol, and simvastatin lactone in L6 cells following acute (0 h) or longer (6 h) exposure by measuring the real-time changes in OCR (Figure 3). Drug concentrations were selected based on those which significantly decreased ATP levels in the metabolic switch assay (Table 3). This method provides a more detailed mechanistic assessment of mitochondrial dysfunction than cellular ATP levels, as several parameters of mitochondrial function are quantified [48,49]. Past research has defined how profiling changes in mitochondrial parameters can be used to classify the specific effect of the drug on ETC: for example, distinguishing between an ETC inhibitor and an ETC uncoupler [43]. Briefly, ETC inhibitors cause a decrease in spare respiratory capacity and/or ATP-linked respiration accompanied by a decrease in BR; ETC uncouplers cause a significant decrease in SRC and/or ALR but with a concomitant increase in BR.

The acute injection of the suspect drugs followed by immediate respirometry changed the mitochondrial profile, but this was not significant when compared to the vehicle control for all drugs, except risperidone (data not shown). However, evidence of mitochondrial toxicity was observed following longer (6 h) exposure to the suspect drugs. Specifically, all tested drugs cause significant decreases in SRC at the highest concentrations tested (Figure 3A–E). Furthermore, all suspect drugs caused a concentration-dependent decrease in ALR, which reached significance at the highest tested concentrations for fenofibrate, pregabalin, risperidone, propofol, and simvastatin lactone. In all cases, there was also a concentration-dependent decrease in BR, which reached significance for all the drugs at the highest concentrations tested. Based on this pattern of changes, these compounds can be classified as direct inhibitors of the electron transport chain.

### 3.4. The Dual Assessment of Cellular ATP Content and Cytotoxicity Can Identify the Early Onset of Mitochondrial Dysfunction before Cell Death in HSKMDC Cells

To provide a preliminary assessment of the translational relevance of findings in the L6 cells to humans, further experiments were performed in a more physiologically relevant human cell model, HSKMDC. Rotenone was again used to confirm the suitability of the cells for the acute metabolic switch assay (Figure 4). In these tests, cellular LDH release was also assessed alongside cellular ATP content to confirm that any change in ATP was not due to cytotoxicity. In HSKMDC, rotenone had an IC_50_-ATPgal of 0.016 ± 0.007 µM, with an IC_50_-ATPglu/gal > 318 (Table 4).

Due to practical limitations, only the two most potent mitotoxic drugs identified in the L6 screen were investigated in HSKMDC: fenofibrate and risperidone. In galactose media, fenofibrate had an IC_50_-ATPgal of 9.7 µM ± 1.1 µM, with an IC_50_-ATPglu/gal > 24; while risperidone had an IC_50_-ATPgal of 143.4 µM ± 51.5 µM, with an IC_50_-ATPglu/gal > 1.9 (Figure 5). With both drugs, the cells remained viable in both media (IC_50_-LDH gal/IC_50_-ATP gal ≥ 2), confirming that the mitochondrial liability preceded cell death.

### 3.5. Respirometry Reveals That Fenofibrate and Risperidone Inhibit Electron Transport Chain Activity in HSKMDC

The acute injection of fenofibrate and risperidone caused a significant concentration-dependent decrease in BR, ALR, and SRC, in contrast to L6 cells, where a 6 h treatment was required for significant decreases to be observed (Figure 6). Each drug reduced ATP-linked respiration, with significant reductions first observed at 3.6 µM of fenofibrate and 11 µM of risperidone (Figure 6). Taken together, based on this pattern of changes, these compounds can be classified as direct inhibitors of the electron transport chain.

## 4. Discussion

Drug-induced rhabdomyolysis is the most serious form of skeletal muscle toxicity. To date, quantitative studies assessing the frequency of drugs that can induce rhabdomyolysis are scarce. However, the availability of such data in FAERS facilitates the identification of these suspect drugs across a large number of reports. Moreover, the FAERS database has universal utility as it complies with international safety reporting guidance, and all reports are coded according to the Medical Dictionary for Regulatory Activities (MedDRA) terminology [50]. Reporting to FAERS is mandatory by law for manufacturers and voluntary for healthcare professionals and consumers. The information is updated regularly by the FDA, as it is used to provide pharmacovigilance information for the safety assessment of suspected drugs, as well as to improve public health [51,52]. Another advantage is its accessibility to the public via the FAERS Electronic Submissions system [53]. In addition, the data/products of interest in the database can be viewed easily as a summary/chart of adverse event reports within a selected period [41]. However, FAERS has important limitations for data analysis. Most importantly, although the FDA dashboard contains valuable information and undergoes careful checking by healthcare providers, it is not possible to assess causality from the report available and whether the reported event is due to the suspected drug, other concomitant drugs, or the disease itself. In addition, the rate of adverse events cannot be verified because it is unlikely that all adverse events or medication errors of a suspect drug have been reported to FAERS. Furthermore, there could be duplicate reports, for example, submitted by the patient and by the company or health care provider. Therefore, every report must be checked for data duplication. Moreover, as data are updated quarterly, the information, incidence rate, etc., may change over time [54].

In total, 27,140 cases of drug-induced rhabdomyolysis were reported in FAERS where the suspect drug had been used alongside other drugs. RORs were used to detect signals of rhabdomyolysis because it is a validated, common disproportionality analysis method to determine a potential association between a medication and an ADR [55,56,57]. Interestingly, the odds of experiencing rhabdomyolysis with fusidic acid were 54.4 (46.3–63.9) times higher than the odds of experiencing this condition with any other drugs. However, in all cases, fusidic acid was reported when administered alongside other drugs and not when used as a monotherapy, which suggests that either this drug does not induce rhabdomyolysis alone, or it may enhance statin-induced rhabdomyolysis, as highlighted in the NHS safety profile, which states that a statin must be stopped when systemic fusidic acid is used as it is associated with fatal rhabdomyolysis [58]. The other drugs with the highest RORs were gemfibrozil, simvastatin, ezetimibe/simvastatin, atorvastatin, fenofibrate, daptomycin, and rosuvastatin, with 15, 10.3, 8, 4.3, 3.8, 3.2, and 2.9 times higher odds, respectively, of rhabdomyolysis than for other drugs. Therefore, mining the case reports deposited in FAERS can lead to observational hypotheses for future potential clinical studies or mechanism-based investigations.

The main objectives of this study were to identify suspected drugs, reported as monotherapy, and then to determine whether drug-induced mitochondrial dysfunction is a common mechanistic cause of drug-induced rhabdomyolysis. By applying further filtering strategies to identify rhabdomyolysis cases arising from monotherapy, a list of 14 suspect drugs was generated. A secondary aim was to assess the utility of murine L6 cells, as a reproducible, cost-effective *in vitro* model to predict muscle toxicity with relevance to humans. L6 cells were selected as they are considered the most suitable model for studies of glucose metabolism and mitochondrial function due to their higher oxygen consumption rate compared with other cell lines [59]. The acute metabolic modification assay is an *in vitro* test that has been widely utilised to successfully identify drug-induced mitochondrial dysfunction in various cell types [47,60,61]. It is based on the ability of most cancer-derived cell lines to generate cellular ATP via both oxidative phosphorylation (OXPHOS) and glycolysis when cultured in glucose-supplemented media, known as the Crabtree effect [62]. Under such conditions, the effects of mitotoxicants on ATP production is masked due to the ability of glycolysis to generate ATP [63]. In the acute metabolic modification assay, the Crabtree effect is exploited to detect compounds that induce mitochondrial toxicity via the direct inhibition of ETC activity. Specifically, cells are forced to rely only on OXPHOS for ATP production by culturing them with an alternative fuel source: galactose [39]. In this system, mitotoxicants that directly inhibit ETC activity cause an early decrease in cellular ATP before any signs of cell death when cultured in galactose, whilst no effects are observed in glucose-supplemented media [64]. Here, we have demonstrated for the first time that L6 cells are amenable to the acute metabolic modification assay using the classic mitochondrial toxin rotenone. In L6 cells, this screen identified that fenofibrate, pregabalin, risperidone, propofol, and simvastatin lactone all induce mitochondrial dysfunction via the direct inhibition of ETC activity, whilst the remaining drugs did not reduce the cellular ATP production. It can be noted that high, supra-clinical concentrations were used in this assay due to the screen’s short timeframe (2 h of exposure). For example, simvastatin lactone is taken over extended periods and the associated skeletal muscle toxicities usually appear within one month of starting the therapeutic regimen [35,65]. However, these unphysiological conditions allow the identification of drug-induced mitochondrial toxicity in the absence of any cell death. Therefore, this initial identification should be followed by further studies to confirm the classification and to assess the potential translational relevance: for example, the high-content screening of mitochondrial membrane potential and respirometry studies under more clinically relevant conditions.

In this study, Seahorse respirometry was used to confirm these results and to provide further mechanistic details, using a previously described methodology to distinguish between drugs that directly inhibit OXPHOS via either the inhibition of ETC or uncoupling of ETC [43,66]. This system is based on the pattern of change of three parameters: BR, ALR, and SRC, following drug exposure. Briefly, a direct inhibitor of complex activity within the ETC would be expected to induce a reduction in BR, ALR, and SRC, whilst an uncoupler of the ETC would induce an increase in BR alongside a concomitant reduction in ALR and SRC. When the cells were treated acutely to investigate the immediate effect of fenofibrate, pregabalin, propofol, or simvastatin lactone, only minor changes occurred in the mitochondrial parameters (data not shown). However, an extended incubation (6 h) allowed for time-dependent effects to be observed, and these results more closely matched those in the acute metabolic switch assays. Fenofibrate, pregabalin, risperidone, propofol, and simvastatin lactone induced a concentration-dependent decrease in BR, ALR, and SRC, thus indicating that they can be grouped with compounds that induce mitochondrial toxicity via the inhibition of ETC. Further, this assay also measures relative potency. Based upon the lowest concentration at which significant decreases in parameters were observed, pregabalin and simvastatin lactone have the most potent inhibitory effect on OXPHOS. In line with previous studies, SRC, which has been described as an early indicator of mitochondrial dysfunction, was the first parameter to be reduced by drug exposure [43].

The literature further confirms the identification of these drugs as mitotoxins using this simple *in vitro* model and assays. Fenofibrate has been reported to cause mitochondrial dysfunction via the inhibition of complex I activity in homogenate rat skeletal muscle [67]. Statins have been shown to induce oxidative stress and activate the mitochondrial apoptosis signalling pathway in isolated muscle biopsies and L6 skeletal muscle [33], whilst another study demonstrates that mitochondria dysfunction associated with statins is due to the impairment of mitochondrial respiration, beta-oxidation, the induction of mitochondrial swelling, cytochrome c release, and DNA fragmentation [68]. Propofol has been shown to impair mitochondrial function via the inhibition of complex II + III activity in an *in vivo* rat model [69]. Moreover, propofol can inhibit fatty acid oxidation and the flux of electrons through the inner mitochondrial membrane, which is more pronounced in human than rat tissue [70] and can attenuate mitochondrial membrane potential in differentiated C2C12 cells [71]. In contrast, pregabalin has not previously been cited as causing mitochondrial dysfunction *in vivo* or *in vitro* skeletal muscle. Furthermore, no studies have reported risperidone to be a mitotoxicant in skeletal muscle. However, several studies have described the interaction of other antipsychotics, aside from risperidone, with the mitochondria in the brain and liver. For instance, studies conducted on mouse and rat brains have observed that certain antipsychotic drugs inhibit complex I of the mitochondrial respiratory chain [72].

It is often stated that rodents are generally less sensitive to mitochondrial toxins, so this may lead to an underestimation of mitotoxicity when using rat-derived L6 cells [73,74]. Therefore, to explore this further, focussed screening assays of two of the drugs identified in the L6 screen, fenofibrate and risperidone, were performed in a more physiologically relevant model, HSKMDC, to explore the translatability of the L6 screening model. Rotenone exposure again confirmed the suitability of these cells for the acute metabolic switch assay. Interestingly, these data confirm this: HSKMDC (IC_50_-ATPgal—0.016 µM) is more sensitive to rotenone than L6 cells (IC_50_-ATPgal—0.036 µM). Both fenofibrate and risperidone were confirmed as mitotoxic in the HSKMDC. However, although HSKMDC was more sensitive to fenofibrate than L6 cells (IC_50_-ATPgal—9.7 µM vs. 30.0 µM, respectively), this was not the case for risperidone (IC_50_-ATPgal—143.4 µM vs. 102.7 µM, respectively). Furthermore, based upon IC_50_-LDHgal/IC_50_-ATPgal, which incorporates a measure of cell death, it is evident that mitochondrial toxicity occurs before cell death, which may be causative. When the effects of the drugs on mitochondria respiration were investigated in HSKMDC, the results again confirmed the findings of the acute metabolic switch assays. Furthermore, the drugs were seen to induce the same pattern of changes, indicating that they should be grouped with inhibitors of ETC activity, as in L6 cells. In addition, the findings in HSKMDC add further support to the utility of L6 cells as a surrogate cell model to identify compounds with the potential to induce mitochondrial toxicity, and perhaps rhabdomyolysis in humans. In addition, the significant reduction in BR, ALR, and SRC occurred at lower concentrations of the test compounds, further supporting the elevated sensitivity of human cells to rat cells to mitotoxicants. The focus of future studies should improve our understanding of the biochemical and molecular mechanisms underlying inter-species variation in sensitivity. Better understanding of species-mediated sensitivity differences will improve the translation to preclinical testing and early identification of compounds with potential for inducing mitochondrial dysfunction in skeletal muscle.

## 5. Conclusions

This study identified 14 drugs that were most commonly reported to induce rhabdomyolysis within FAERS as monotherapy, and five of these were further identified as mitochondrial toxicants via a direct effect on OXPHOS. Overall, the study demonstrates that the L6 model of acute modification may find utility as an initial, cost-effective screen for identifying potential myotoxicants with relevance to humans and, importantly, that drug-induced mitochondrial dysfunction may be a common mechanism shared by some drugs that induce myotoxicity. However, this should only act as a first tier of testing, with any suspected compounds undergoing further investigation to define the mechanism of toxicity in a primary human cell model, such as HSKMDC. Overall, these findings have the potential to aid drug developers to identify compounds with the potential to induce skeletal muscle adverse events early in the drug development process, as well as help scientific researchers to investigate the mechanisms of drug-induced muscle toxicity and to understand why certain individuals are more susceptible to such toxicity. In conclusion, these studies suggest that drug-induced mitochondrial dysfunction may be linked to the onset of severe skeletal muscle toxicity.

## Figures and Tables

**Figure 1 biomedicines-11-01485-f001:**
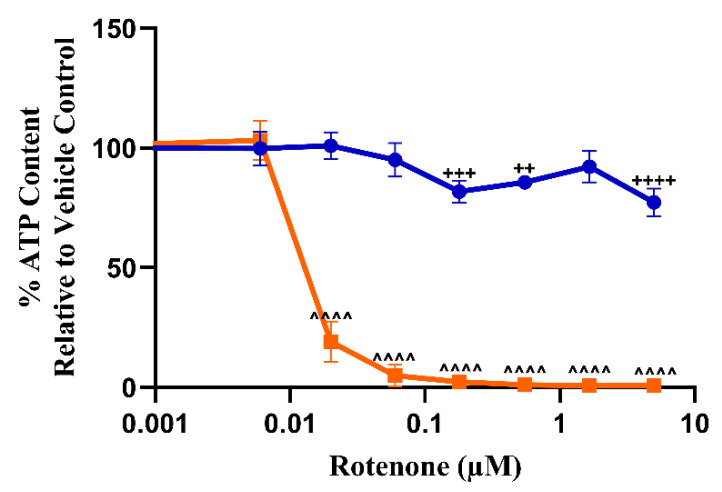
The effect of rotenone (0–5 µM, positive control) exposure on ATP levels of L6 cells (2 h), compared to the vehicle control. L6 cells were exposed to rotenone in glucose or galactose serum-free media. Statistical significance compared to the vehicle control was determined by one-way ANOVA with Dunnett’s correction for multiple comparisons. ATP glucose ^++^
*p*-value < 0.01, ^+++^
*p*-value < 0.001, ^++++^
*p*-value < 0.0001. ATP galactose ^^^^ *p*-value < 0.0001. ATP levels are reported as the percentage of the vehicle control ± S.D. (*n* = 3). Key: blue circles represent ATP glucose, and orange squares represent ATP galactose.

**Figure 2 biomedicines-11-01485-f002:**
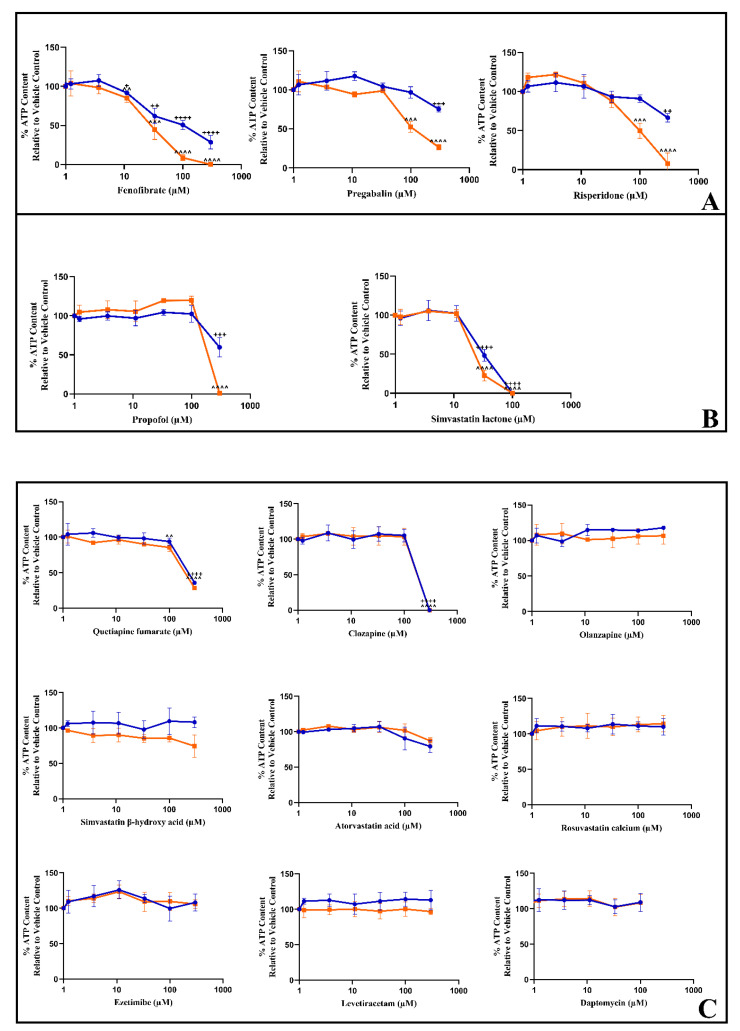
The effect of the test compounds on the cellular ATP content of L6 cells (2 h) compared to the vehicle control. Serial concentrations of the compounds were used up to 300 µM (or 100µM for SVL and DAP) in glucose or galactose serum-free media. (**A**) Compounds were labelled as having a mitochondrial liability. (**B**) Compounds displayed a multi-mechanistic reduction in ATP. (**C**) Compounds were negative for mitochondrial toxicity. Values are displayed as mean ± S.D. (*n* = 3). Statistical significance compared to the vehicle control was determined by one-way ANOVA with Dunnett’s correction for multiple comparisons. ATP glucose ^+^ *p*-value < 0.05, ^++^
*p*-value < 0.01, ^+++^
*p*-value < 0.001, ^++++^
*p*-value < 0.001. ATP galactose ^^ *p*-value < 0.01, ^^^ *p*-value < 0.001, ^^^^ *p*-value < 0.0001. ATP levels are reported as the percentage of the vehicle control ± S.D. (*n* = 3). Key: blue circles represent ATP glucose, and orange squares represent ATP galactose.

**Figure 3 biomedicines-11-01485-f003:**
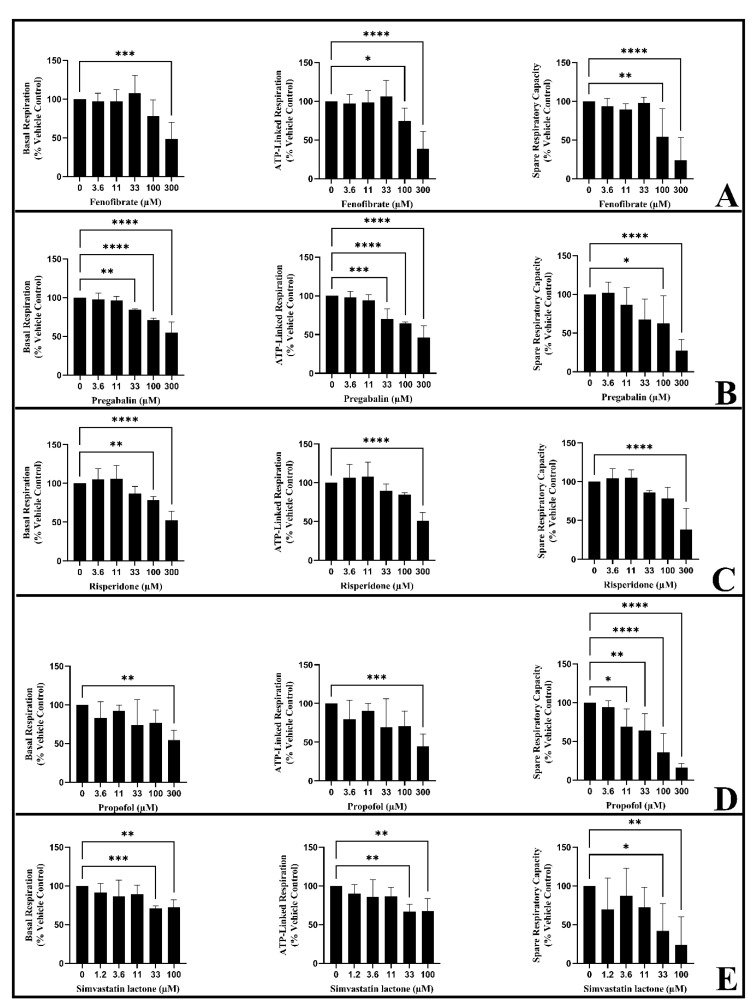
Examining the effect of the suspect drugs (6 h) on respiratory parameters in L6 cells. (**A**) fenofibrate, (**B**) pregabalin, (**C**) risperidone, (**D**) propofol, (**E**) simvastatin lactone. Values are expressed as a percentage of vehicle control and expressed as mean ± S.D. (*n* = 3). Statistical significance compared to the vehicle control was determined by two-way ANOVA with Dunnett’s correction for multiple comparisons. * *p*-value < 0.05, ** *p*-value < 0.01, *** *p*-value < 0.001, **** *p*-value < 0.0001.

**Figure 4 biomedicines-11-01485-f004:**
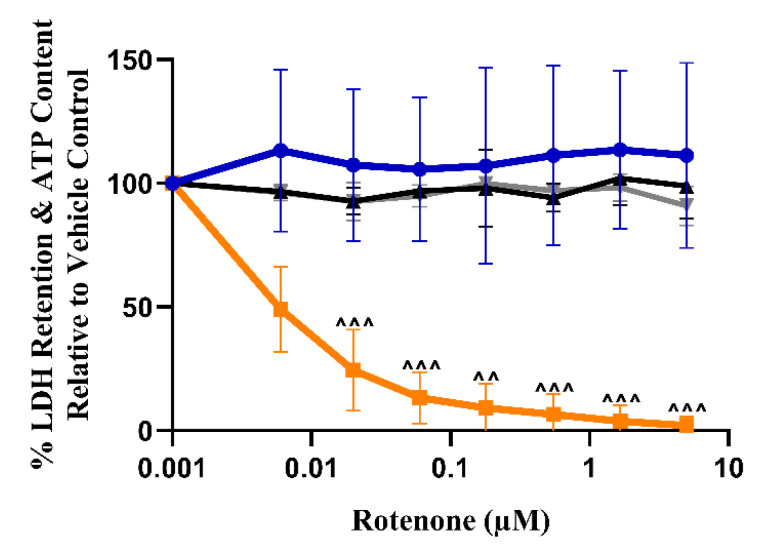
The effect of rotenone (0–5 µM, positive control) exposure on ATP levels and LDH retention in HSKMDC (2 h), compared to the vehicle control. HSKMDC cells were exposed to rotenone in glucose or galactose serum-free media. Statistical significance compared to the vehicle control was determined by one-way ANOVA with Dunnett’s correction for multiple comparisons. ATP galactose ^^ *p*-value < 0.01, ^^^ *p*-value < 0.001. ATP levels are reported as the percentage of the vehicle control and expressed as mean ± S.D. (*n* = 3). Key: blue circles represent ATP glucose, orange squares represent ATP galactose, the black triangle represents cytotoxicity-HSKMDC–glucose, and the grey triangle down represents cytotoxicity–HSKMDC–galactose.

**Figure 5 biomedicines-11-01485-f005:**
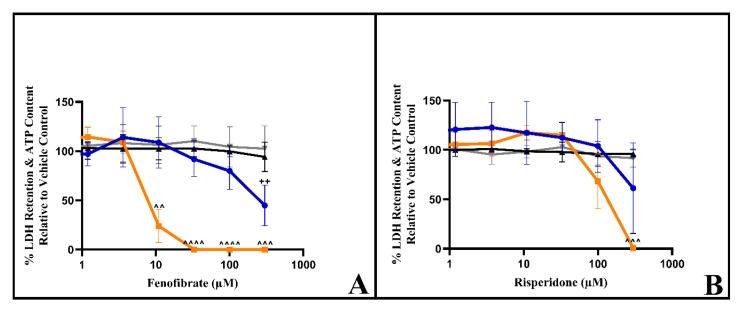
The effect of (**A**) fenofibrate and (**B**) risperidone exposure (2 h) on ATP content and LDH retention in HSKMDC. Serial concentrations of the compounds were used up to 300 µM in glucose or galactose serum-free media. ATP and cytotoxicity levels are expressed as a percentage of the corresponding media vehicle control and values are displayed as mean ± S.D. (*n* = 3). Statistical significance compared to the vehicle control was determined by one-way ANOVA with Dunnett’s correction for multiple comparisons. ATP glucose ^++^
*p*-value < 0.01. ATP galactose ^^ *p*-value < 0.01, ^^^ *p*-value < 0.001, ^^^^ *p*-value < 0.001. Key: blue circles represent ATP glucose, orange squares represent ATP galactose, the black triangle represents cytotoxicity-HSKMDC–glucose, and the grey triangle down represents cytotoxicity–HSKMDC–galactose.

**Figure 6 biomedicines-11-01485-f006:**
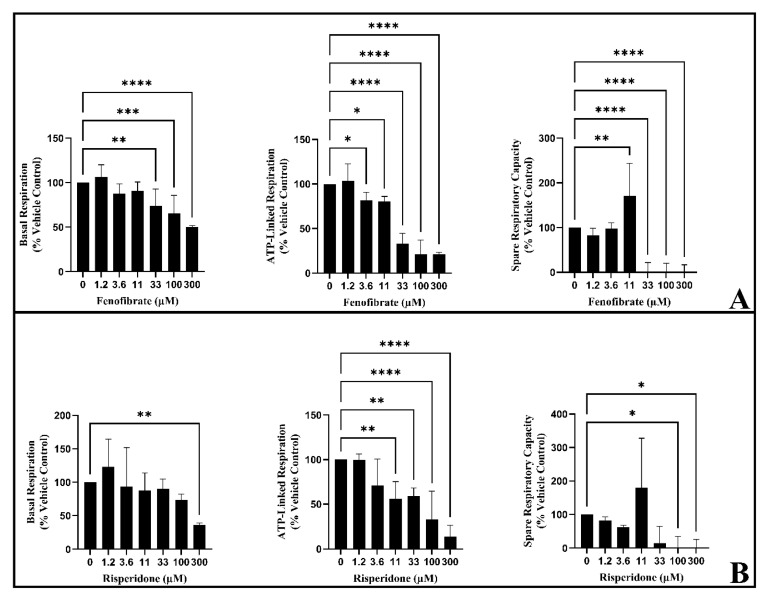
Examining the effect of the acute injection of (**A**) fenofibrate and (**B**) risperidone on respiratory parameters in HSKMDC. Values are expressed as a percentage of vehicle control ± S.D. (*n* = 3). The values were normalised to μg of protein per well. Statistical significance was determined by two-way analysis of variance (ANOVA) with Dunnett’s correction for multiple comparisons. * *p*-value < 0.05, ** *p*-value < 0.01, *** *p*-value < 0.001, **** *p*-value < 0.001.

**Table 1 biomedicines-11-01485-t001:** The most common drugs reported to FAERS suspected to induce rhabdomyolysis during the period 2004 to 31 March 2020. There were 2153 suspected drugs with a total of 27,140 rhabdomyolysis cases, of which statins were most heavily reported. Indeed, cerivastatin was withdrawn from the market due to increased reports of rhabdomyolysis [44]. This table represents the top 31 suspect drugs, with the percentages based upon those which represent greater than 1% of total rhabdomyolysis cases. Some cases were reported with more than one suspected drug.

Suspected Drug	Number of Rhabdomyolysis Cases	(%)	ROR	95% CI	Suspected Drug	Number of Rhabdomyolysis Cases	(%)	ROR	95% CI
Simvastatin	4705	17.3	10.3	10.0–10.7	Gemfibrozil	442	1.6	15.0	13.5–16.6
Atorvastatin	3307	12.2	4.3	4.1–4.4	Propofol	434	1.6	2.2	2.0–2.4
Rosuvastatin	2008	7.4	2.9	2.7–3.0	Clarithromycin	427	1.6	1.2	1.1–1.4
Quetiapine	761	2.8	0.5	0.5–0.6	Venlafaxine	425	1.6	0.5	0.5–0.6
Risperidone	654	2.4	0.5	0.5–0.6	Aripiprazole	419	1.5	0.4	0.4–0.4
Olanzapine	648	2.4	0.9	0.8–0.9	Amlodipine	405	1.5	0.5	0.5–0.6
Levetiracetam	588	2.2	0.9	0.9–1.0	Daptomycin	380	1.4	3.2	2.9–3.6
Ezetimibe/Simvastatin	541	2.0	8.0	7.3–8.8	Omeprazole	363	1.3	0.5	0.4–0.5
Cyclosporine	505	1.9	0.5	0.5–0.6	Sertraline	347	1.3	0.4	0.3–0.4
Ezetimibe	501	1.9	2.6	2.4–2.9	Mirtazapine	318	1.2	0.8	0.7–0.9
Furosemide	480	1.8	0.8	0.7–0.9	Lamotrigine	314	1.2	0.3	0.3–0.4
Metformin	478	1.8	0.4	0.4–0.4	Aspirin	313	1.2	0.2	0.2–0.2
Acetaminophen	449	1.7	0.3	0.3–0.3	Haloperidol	310	1.1	1.3	1.1–1.4
Fenofibrate	449	1.7	3.8	3.4–4.2	Alprazolam	290	1.1	0.4	0.4–0.5
Pregabalin	442	1.6	0.2	0.2–0.3	Clozapine	286	1.1	0.2	0.2–0.3
					Fusidic Acid	285	1.1	54.4	46.3–63.9

Abbreviations: ROR, reported odds ratio; CI, 95% confidence intervals.

**Table 2 biomedicines-11-01485-t002:** The drugs in FAERS that are most frequently reported to be suspected of causing rhabdomyolysis as a monotherapy. Data are presented from the period 2004 to 31 March 2020. Statins were associated with the most reports of rhabdomyolysis representing 65.7% of cases within the top 14 drugs. Among those reported were 576 cases (8.7%) with fatal outcomes.

Suspected Drug Reported as a Single Drug Used	Number of Rhabdomyolysis Cases (*n* = 6583)	(%)	Number of Death Cases (*n* = 576)
Simvastatin	1815	27.6	161
Atorvastatin	1386	21.1	169
Rosuvastatin	1122	17.0	47
Levetiracetam	386	5.9	8
Ezetimibe/Simvastatin	350	5.3	10
Olanzapine	258	3.9	23
Daptomycin	227	3.5	18
Quetiapine	199	3.0	26
Propofol	179	2.7	70
Ezetimibe	166	2.5	6
Pregabalin	132	2.0	13
Fenofibrate	126	1.9	6
Risperidone	121	1.8	7
Clozapine	116	1.8	12

**Table 3 biomedicines-11-01485-t003:** Table reporting IC_50_ values for each suspect drug tested in L6 cells using the acute metabolic switch assay. Values were determined by fit spline and interpolating IC_50_ concentrations from the standard curve. Results are displayed as mean ± S.D. (*n* = 3). Statistical significance was determined by unpaired *t*-test with Welch’s correction. Abbreviations: ns, not significant; n/d, the value could not be determined.

	ATP IC_50_ (μM) ± S.D. Glucose Galactose		IC_50_-ATPglu/IC_50_-ATPgal (*p*-Value)
Control compound: Rotenone	>5	0.036 ± 0.008	>138.9 (<0.0001)
Tested reported drugs: A. Compounds were found to be positive for mitotoxicity following 2 h exposure in L6 cells (IC_50_-ATPglu/IC_50_-ATPgal ≥ 2).			
Fenofibrate	95.8 ± 27.6	30.0 ± 19.4	3.2 (0.0330)
Pregabalin	>300	116.0 ± 2 2.5	>2.6 (0.0049)
Risperidone	>261.0 ± 67.5	102.7 ± 28.2	>2.5 (0.0401)
B. Compounds were found to have a multi-mechanistic mechanism for mitotoxicity following 2 h exposure in L6 cells (IC_50_-ATPglu/IC_50_-ATPgal < 2).			
Propofol	>300	241.6 ± 13.4	>1.2 (0.0172)
Simvastatin Lactone	40.8 ± 20.7	28.1 ± 9.4	1.5 (0.4116)
C. Compounds were found to be negative for mitotoxicity following 2 h exposure in L6 cells (IC_50_-ATPglu/IC_50_-ATPgal = 1).			
Quetiapine	265.2 ± 4.9	216.5 ± 57.8	1.2 (0.2814) (ns)
Clozapine	218.3 ± 13.2	213.2 ± 10.1	1 (0.6270) (ns)
Olanzapine	>300	>300	~1 (n/d)
Simvastatin acid	>300	>300	~1 (n/d)
Atorvastatin acid	>300	>300	~1 (n/d)
Rosuvastatin	>300	>300	~1 (n/d)
Ezetimibe	>300	>300	~1 (n/d)
Levetiracetam	>300	>300	~1 (n/d)
Daptomycin	>100	>100	~1 (n/d)

**Table 4 biomedicines-11-01485-t004:** Table reporting IC_50_ values for each suspect drug tested in HSKMDC using the acute metabolic switch assay. Values were determined by fit spline and interpolating IC_50_ concentrations from the standard curve. Results are displayed as mean ± S.D. (*n* = 3). Statistical significance was determined by unpaired *t*-test with Welch’s correction.

	ATP IC_50_ (μM) ± S.D. Glucose Galactose		IC_50_-ATPglu/ IC_50_-ATPgal (*p* Value)	LDH IC_50_ (μM) ± S.D. Glucose Galactose		IC_50_-LDH gal/ IC_50_-ATP al
Rotenone	>5	0.016 ± 0.007	>318.4 (<0.0001)	>5	>5	>312.5
Fenofibrate	>300	9.7 ± 1.1	>24.0 (0.028)	>300	>300	>30.9
Risperidone	>300	143.4 ± 51.5	>1.9 (0.0419)	>300	>300	>2.1

## Data Availability

All data are available upon reasonable request from the authors.

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
