# Peer review of "Developing In Vitro Models to Define the Role of Direct Mitochondrial Toxicity in Frequently Reported Drug-Induced Rhabdomyolysis"

_biomedicines, 2023, doi:10.3390/biomedicines11051485_

Round 1

Reviewer 1 Report

The article is well written. The research methodology was described in detail. The subject of the work concerns an issue important from a medical point of view, touching on the issue of life-threatening rhabdomyolysis.
The analyses carried out by the authors focused on the issue of mitotoxicity of selected drugs. Such information is undoubtedly important for clinicians, the pharmaceutical market, and the patient.
The authors showed the usefulness of the model based on L6 cells in mitotoxicity studies. However, this model has been used before in drug research.
As the authors themselves describe, in the case of most compounds, their effect on mitochondria has already been described, in most cases also in muscle cells.
Articles on rhabdomyolysis and the drugs are also described in the literature.
The authors also do not explain why fenofibrate and risperidone were chosen for HSKMDC studies.
Why, for example, pregabalin, which has not been described due to authors in the context of mitochondrial dysfunction in skeletal muscles, was not included in these studies.
However, in the article of Jiang W. et al. (2016) the few published cases related to levetiracetam, phenytoin, valproic acid lamotrigine, gabapentin, and pregabalin were collected.
It has also long been known that cells, such as muscle cells, whose energy requirements are very high, are sensitive to factors that damage the mitochondria, and both clinicians and drug developers are well aware of this.
From this point of view, the article is not very innovative and is essentially limited to confirming the usefulness of L6 cells in mitotoxicity studies.

Author Response

Thank you so much

Reviewer 2 Report

The topic of this manuscript is of large general interest as rhabdomyolysis is an adverse effect of several drugs and it is a severe one. Numerous approaches have been made to elucidate the mechanisms causing rhabdomyolysis. Here, two cellular models are used to quantitatively compare the in vitro effects of a choice of the worst drugs.

Comments
The text of this manuscript is on one hand wordy and on the other hand insufficiently precise. The same results are often repeated. The sentence “Briefly, rotenone is an exemplar inhibitor of ETC (complex I inhibitor) and it reduces the BR, ALR, and SRC; whilst 2-[2-(3-Chlorophenyl) hydrazinylyidene] propanedinitrile (CCCP), is a potent exemplar mitochondrial oxidative phosphorylation uncoupler, which induces an increase the BR, with a concomitant reduction in ALR and SRC [61, 62].“ is an example of a wordy one and also contains two typos.
Some rather superficial explanations are given instead of detailed mechanistic interpretations. Similarly, not enough scientific reflection has gone into the discussion.

The following are examples of problems but there are more of them:

1.     In Fig. 6, the values are about twice above baseline at 11 micromolar for both drugs. It is stated “Interestingly, both drugs at 11 μM show a significant increase in SRC, which could be interpreted as a compensatory mechanism activated as a cellular defence. » First, the value for risperidone (Fig. 6B) is not significant. Second, the interpretation is rather a speculation. Why should a cellular defence mechanism only take place at one single concentration? This eye-catching result is strange and needs to be explained. It could also be that there was rather sample handling problem or perhaps n=3 is insufficient.

2.     Why is “rhabdomyolysis a large cost for the pharmaceutical industry?

3.     Important: what is the effect of 1 micromolar dexamethasone (line 103 0.4 μg/mL) on mitochondrial effects alone and in the presence of the test compounds? Note that dexamethasone has a nM effect on the glucocorticoid receptor.

4.     Fig 1 & 2: What were the compounds besides rotenone? (“Serial dilutions of the compounds …”). It is not clear why “Black circles” and “Black squares” mean as the lines are in color. This is a trivial error – it indicates to the reviewer that insufficient care has gone into this manuscript.

5.     There is nothing under:
Author Contributions:
Funding:

Institutional Review Board Statement:
Informed Consent Statement:
Data Availability Statement:
Conflicts of Interest:

Minor comments
Line 63 – “Firstly in a cost-practical screen was undertaken, using L6 rat-derived skeletal muscle cells, …” this shows to me that proof reading was insufficient.
Line 69 – what is “VA, USA”? similar for CA, USA
Line 70 – why …”forms of” …
Line 89 – why is generic name in “?
Line 109 – what is “
0.25 % trypsin-EDTA » ?
Tables 1 & 2: the mention of the salts « calcium, hydrochloride » is not appropriate.
References: Names of Journals are not correctly abbreviated, e.g. refs. 8, 9,23,25,26,43,51,63,69,71.
refs. 44 & 45 are misleading
Line 578 Ref. 8 The first names are given as last names. Gianfranco Ivan Guiseppe
Lines 381,382, 421, 541 – no comma
Lines 441-447 – these sentences need to be improved

Line 415 remove “a list of”
Line 434 remove “
this switch is” or put a ; before
Lines 470-474 – see above
Lines 475 and 479, 484 – it is not dose but concentration
Line 500 – sentence “
With another study demonstrates that …”
Line 507 – “…a
nd can attenuated mitochondrial membrane potential
Line 529 – “
Seahorse respirometry confirmed the acute metabolic switch assays » the results were confirmed.

Author Response

Thank you so much.
